# Deduced Respiratory Scores on COVID-19 Patients Learning from Exertion-Induced Dyspnea

**DOI:** 10.3390/s23104733

**Published:** 2023-05-13

**Authors:** Zijing Zhang, Jianlin Zhou, Thomas B. Conroy, Samuel Chung, Justin Choi, Patrick Chau, Daniel B. Green, Ana C. Krieger, Edwin C. Kan

**Affiliations:** 1School of Electrical and Computer Engineering, Cornell University, Ithaca, NY 14853, USA; 2Center for Sleep Medicine at Weill Cornell Medicine, New York, NY 10065, USA

**Keywords:** clinical diagnosis, COVID-19, dyspnea, noninvasive, pulmonary disorders, respiratory monitoring

## Abstract

Dyspnea is one of the most common symptoms of many respiratory diseases, including COVID-19. Clinical assessment of dyspnea relies mainly on self-reporting, which contains subjective biases and is problematic for frequent inquiries. This study aims to determine if a respiratory score in COVID-19 patients can be assessed using a wearable sensor and if this score can be deduced from a learning model based on physiologically induced dyspnea in healthy subjects. Noninvasive wearable respiratory sensors were employed to retrieve continuous respiratory characteristics with user comfort and convenience. Overnight respiratory waveforms were collected on 12 COVID-19 patients, and a benchmark on 13 healthy subjects with exertion-induced dyspnea was also performed for blind comparison. The learning model was built from the self-reported respiratory features of 32 healthy subjects under exertion and airway blockage. A high similarity between respiratory features in COVID-19 patients and physiologically induced dyspnea in healthy subjects was observed. Learning from our previous dyspnea model of healthy subjects, we deduced that COVID-19 patients have consistently highly correlated respiratory scores in comparison with normal breathing of healthy subjects. We also performed a continuous assessment of the patient’s respiratory scores for 12–16 h. This study offers a useful system for the symptomatic evaluation of patients with active or chronic respiratory disorders, especially the patient population that refuses to cooperate or cannot communicate due to deterioration or loss of cognitive functions. The proposed system can help identify dyspneic exacerbation, leading to early intervention and possible outcome improvement. Our approach can be potentially applied to other pulmonary disorders, such as asthma, emphysema, and other types of pneumonia.

## 1. Introduction

Dyspnea, also known as the patient’s feeling of difficult or labored breathing, is a clinical symptom nearly as important as pain, affecting a quarter of the general population and half of seriously ill patients [1,2]. Dyspnea can be a prevalent manifestation in conditions such as chronic obstructive pulmonary disease (COPD), bronchitis, asthma, coronavirus disease 2019 (COVID-19), pneumonia, heart failure, and panic disorders [3]. Dyspnea can be further divided into acute onset and chronic dyspnea. The latter, by definition, is present for more than four weeks. COVID-19 caused by severe acute respiratory syndrome coronavirus-2 (SARS-CoV-2) has rapidly spread across the globe since 2020. Over 30% of patients with COVID-19 have experienced chronic dyspnea [4].

Dyspnea typically sets in between the fourth and eighth day of illness. The timing of dyspnea may be one of the most important hallmarks of more severe COVID-19 infection, especially for clinicians seeing patients in an ambulatory setting [5]. Studies found that dyspnea, rather than fever [6], was significantly associated with higher mortality in COVID-19 patients [7,8]. The initial days after the onset of dyspnea are critical for identifying progressively worsening conditions [1]. In addition to the dyspnea experienced during the acute phase of COVID-19 infection, dyspnea may also be found in association with the long-term sequelae post-COVID-19, or so-called long COVID-19, which is thought to affect 10–50% of COVID-19 survivors [9]. Dyspnea is also a frequent symptom of post-COVID-19 syndrome (PCS) [10,11]. Some patients with persistent dyspnea after recovering from COVID-19 have also reported a decrease and/or dysfunction in myocardial performance [12,13].

The dyspnea sensation for COVID-19 patients is often derived from the decreased ventilation efficiency caused by pneumonia or related bronchitis. A distinct phenotype in long COVID-19 is that patients have reduced exercise tolerance and experience exertional dyspnea more easily, even though major pulmonary parenchymal and airway abnormalities can no longer be identified with chest imaging [14,15].

In present clinical practices, dyspnea is mainly assessed by self-reports from patients. Dyspnea scores can be assessed in person or remotely [16] using patient interviews, and augmented by subjective measures, such as the Medical Research Council (MRC) Dyspnea Scale [17] and Borg visual analog scale (VAS) [18]. Studies indicated that subjective dyspnea measures have inadequate accuracy in high-risk COVID-19 patients, not only because the sensation is gradual and varies with time but also because the patients can become nervous after knowing positive test results [19], both of which can contribute to biases in the self-report. The subjective dyspnea score can also vary for each person based on emotion and tolerance and can be challenging to assess for those who refuse to cooperate or cannot communicate due to medical issues, such as stroke, dementia, and loss of speech. Frequent queries to patients for continuous dyspnea evaluation are tedious and stressful, especially over the sleep period, and hence less practical for timely prognosis and diagnosis.

Dyspnea is highly correlated with pulmonary functions [20]. Surrogate measures of respiratory function can indirectly help determine dyspnea severity, but existing techniques have limitations. Pulmonary function tests can only capture respiratory measures at discrete points in time [21] and require adequate patient effort and cooperation. Chest computed tomography (CT) can provide high-resolution images of the lungs, but it is expensive, requiring specialized equipment that is often unavailable, producing only discrete lung snapshots in a dedicated clinical setup and providing limited utility for continuous assessment of lung function [22]. Respiratory inductance plethysmography (RIP) [23], strain gauge (SG) [24], and spirometer [25] measure lung function but can be uncomfortable, requiring connection to immobile machines, operator assistance, and patient cooperation. Thus, they are not often feasible for repetitive or continuous monitoring.

Considering the limitations of the current methods to monitor respiratory function and assess dyspnea, we propose using a noninvasive and wearable radio-frequency (RF) respiratory sensor and a machine-learning (ML) model to provide correlated respiratory scores for COVID-19 patients, augmented by our previous study on the objective dyspnea scoring system [26], which was based on the correlation between continuous respiratory features and self-reported dyspnea scores during physiologically induced dyspnea episodes.

We collected overnight clinical respiratory data of patients admitted to the hospital due to acute COVID-19 (N = 12) from the Weill Cornell Center for Sleep Medicine. Self-reported dyspnea scores were only collected at admission as frequent inquires afterward can be tiresome with aggravated biases. These patients also confirmed pulmonary involvement based on radiological imaging. To benchmark the results, we also performed a separate human study (N = 13) on healthy participants using exactly the same experimental setup. We analyzed the statistical distribution of respiratory metrics from COVID-19 patients and healthy controls and demonstrated a high similarity between respiratory features in COVID-19 patients and physiologically induced dyspnea in healthy subjects. By training on our previous objective dyspnea scoring model (N = 32) on healthy subjects with induced dyspnea from exertion and airway blockage [26], we can further produce continuous “deduced respiratory scores” of COVID-19 patients using wearable sensors to supplement self-reported dyspneic episodes. In the future, our respiratory sensors and the deduced respiratory scoring system can be potentially expanded to surrogate dyspnea evaluation in patients with asthma, COPD, and other types of pneumonia.

## 2. Materials and Methods

### 2.1. Experimental Design and Data Processing

#### 2.1.1. Experimental Setup

To monitor the respiration in hospitalized COVID-19 patients with confirmed lung infection by chest imaging, we built an all-in-one wearable RF near-field coherent sensing (NCS) unit [27,28] on a 4-layer printed circuit board (PCB), as shown in Figure 1a. The block diagram of the entire sensing unit is shown in Figure 1b. Two SimpleLink modules (Texas Instrument CC1310) with sub-1 GHz ultra-low-power microcontrollers were used as the sensing transmitter (Tx) and receiver (Rx). The transceivers employed quadrature I/Q modulation, where two channels of 12-bit, I and Q samples, were sampled at 2 kHz and accumulated into one 400-byte cyclic buffer at the sensor Rx. Once the I/Q buffer was filled, the Rx radio core would bundle the I and Q samples with the readings of temperature, accelerometer, and gyroscope to a micro secure digital (SD) card through the serial peripheral bus (SPI). The temperature sensor (Texas Instrument TMP112, Austin, TX, USA) and inertial measurement unit (Bosch Sensortec BMI160, Reutlingen, Germany) were connected by the inter-integrated circuits (I2C) serial protocol. The battery provided the system power through a low-dropout (LDO) module.

#### 2.1.2. Data Collection from COVID-19 Patients

Respiratory data acquisition from COVID-19 patients took place at Weill Cornell Medicine between July 2021 and March 2022. Patients admitted to New York Presbyterian Hospital with COVID-19 symptoms and lung imaging confirmation were offered participation in the study. All participants signed an informed consent form. The study protocol was reviewed and approved by the Weill Medical Center Institutional Review Board (IRB Protocol #: 20-06022181).

Upon enrollment, patient demographic, health status, and baseline vital-sign data were gathered, as shown in Table 1a. Age information was not gathered for the COVID-19 dataset. Health status information included hypertension, obstructive sleep apnea, cancer history, asthma, COPD, and other chronic lung diseases. Baseline vital-sign data included heart rate (HR), breath rate (BR), temperature, and oximetry oxygen saturation (SpO2). Other information included medications administered or the presence of ventilation or supplemental oxygen during recording. After the admission information was gathered, the medical staff applied the ApneaLink device (Resmed^®^, San Diego, CA, USA) with two NCS sensing units on the patient’s chest for overnight monitoring. The two sensors were synchronously powered on to begin recording. Patients wore the sensor for an average of 14.3 h overnight. The demographic distribution of COVID-19 patients is shown in Appendix A.

#### 2.1.3. Healthy Participant Study Protocol

To further investigate the correlation between the respiratory score of COVID-19 patients and physiologically induced dyspnea in healthy subjects, we conducted another human study (the results are shown in in Table 1a (Exp 1)) on 13 healthy participants with self-reported dyspnea scores and respiratory waveform measurements when dyspnea was induced by exercise. For fair comparison without concerns about hardware difference, we used the same wearable respiratory sensors as in the COVID-19 data collection. Figure 1c shows the experimental setup with the participant wearing two sensors on the left and on the right. The vertical position of the two sensors is at the level of the sternum, roughly between the 3rd and 7th ribs. The demographic distribution of 13 healthy participants is shown in Appendix A. The human study was approved by Cornell Institutional Review Board (IRB) Protocol ID #1812008488. Written informed consent was obtained from all participants. The participant was first asked to sit in a chair in a relaxed position for normal breathing for 30 min. To induce dyspnea, the participant would follow a 5 min cardio exercise video [29] and then sit back in the chair, and the participant’s post-exercise breathing would be recorded for 5 min. The participants were asked to report subjective dyspnea scores *D_self_* several times on the Borg scale, as shown in Figure 1f [18], during the transition points of the study. The Borg scale is widely used in clinical assessment of dyspnea: 0 represents no dyspnea sensation at all, while 10 indicates the maximum level of dyspnea.

As shown in Table 1a, we also adopted our previous dyspnea study for model training and comparison [26], denoted as data from Exp 2, where we utilized the software-defined radio (SDR, Ettus B210) for the RF sensor, as shown in Figure 1d,e. Two wired NCS sensors were placed on the chest and the abdomen in the front torso. In this human study, participants first recorded 5 min normal breathing sitting in a chair, then used aerobic exercise to introduce dyspnea, and then recorded another 5 min of post-exercise breathing. The dyspnea was also induced through a face mask to change the respiratory resistance. The Borg dyspnea score was reported several times throughout the routine. Figure 2 presents examples of respiratory waveforms we acquired from different datasets. Y-axes are individually normalized in different channels. When multiple datasets are available, we would choose the waveforms with the most consistent features, as the comparison and learning models are all based on extracted respiratory features, and unstable waveform details are less important. For the COVID-19 dataset, we most often utilized the accelerometer channel for respiration monitoring. For Exp 1, we demonstrated NCS and accelerometer channels with consistent features for both normal and exertion modes. For Exp 2, we demonstrated NCS recording for the same routines. Comparing normal breathing in Figure 2b,d with exertion in Figure 2c,e for both Exp 1 and Exp 2, we can observe a distinct change in breath rates (BR), with an increase in rate and a decrease in breath-to-breath variation after exertion. We can also observe that COVID-19 patients had higher BR than healthy participants during normal breathing, and the COVID-19 waveforms were similar in frequency and amplitude to those acquired after exertion in healthy participants. Respiratory parameters were extracted from the waveforms after min-max peak detection [26] as shown in Figure 2f, including inter-breathing intervals (IBI), inspiration intervals (IN), expiration intervals (EX), and peak-to-peak magnitude (PP).

### 2.2. Statistical Analysis

#### 2.2.1. Data Preprocessing

After gathering data from healthy participants and COVID-19 patients, we preprocessed our datasets and then extracted the respiratory features to be used in the ML algorithms for dyspnea evaluation and deduced respiratory scoring. We used MATLAB for signal processing and feature extraction and Python for ML algorithms.

For datasets in COVID-19 and Exp 1 described in Table 1a, we retrieved respiratory waveforms from 1 NCS (amplitude) and 6 accelerometer (translational and rotational) channels. For Exp 2, we utilized multiple NCS channels from the thorax and abdomen. Different channels and datasets went through the same signal-processing procedure for a fair comparison. As shown in Figure 3, we first down-sampled all datasets to 20 Hz, and then bandpass-filtered waveforms from 0.05 Hz to 2 Hz to remove the DC drift and high-frequency noises. Savitzky–Golay 4th-order finite impulse response (FIR) smoothing filter [30] was further employed to perform a local polynomial regression to smooth the waveform. For the long recording in each dataset, we opted to segment waveforms into short epochs of *T_epoch_* = 60 s with a sliding window of *T_slide_* = 30 s for feature extraction.

After epoch segmentation, waveforms were normalized to center at 0 with a standard deviation of 1.0 in each epoch for every channel. Then we extracted features in each epoch for data analysis and constructed the ML model in the next section. We implemented the peak detection algorithm [31] by tracing a constantly updated moving-average curve in a given window. Then local maxima and minima were accordingly labeled for parameter extraction. An example is shown in Figure 2f, where the green line is the filtered respiratory waveform from a COVID-19 patient, and the red and yellow triangles mark the maximum and minimum peaks detected by the moving-average algorithm. The false peaks caused by the noise were mostly avoided. Respiratory parameters in each breath cycle were extracted to represent the instantaneous respiratory characteristics, as shown in Table 2a.

After gathering respiratory cycles and parameters, we extracted 37 respiratory features as shown in Table 2b. The first three features were: (1) mean (*µ*); (2) standard deviation (*σ*); (3) coefficient of variation (*CoV*) of the respiratory parameters in Table 2b, where *CoV* was defined as
(1)$$CoV=({\frac{\sigma }{\mu })}^{2}$$
showing the extent of variability in relation to the mean.

To further capture variability between adjacent breaths, *ℜ* was the autocorrelation in a time lag of one respiratory cycle to measure the successive similarity of the given respiratory parameter. Representing the successive difference, *ς* was defined as the mean absolute difference between adjacent cycles. Additionally, Skew and kurt measured the tailedness and asymmetry of each respiratory cycle and were averaged over all cycles within the epoch. Cycle denoted the total number of detected respiratory cycles in the epoch, and entropy denoted the total randomness or entropy of the waveform.

Apart from 37 respiratory features extracted from the time domain, we added 10 features extracted from the frequency domain as shown in Table 2c. The parameters *ƞ_fi_* and *ƿ_f1_* (*i* = 1~4) represented the power in specific bandwidth divided by the total power in all frequencies and time-averaged power density (dB/Hz), respectively. The five chosen bandwidths were *f_1_* = (0, 0.4) Hz (the main breathing frequency range); *f_2_* = (0.4, 1) Hz; *f_3_* = (1, 2) Hz; *f_4_* = (*f_BR_* − 0.15, *f_BR_* + 0.15) Hz. The bandwidth *f_BR_* was first estimated from the average BR provided by hospital reports for every patient, and then further refined to be the local BR in every epoch by finding the maximal energy in the possible frequency band. Signal-to-noise ratios (SNR) in BR were denoted by *SNR_BR_* which were calculated by the maximal energy on the *f_BR_* divided by the estimated noise power.

#### 2.2.2. Channel and Epoch Selection

After segmentation and feature extraction, we selected the optimal channel and epoch from the datasets according to the estimated signal quality. For the accelerometer, we had 6 channels consisting of X, Y, and Z translational and rotational motions, as shown in Figure 4 for an example. Feature extraction was performed on every channel, and the optimal channel was decided by the least variation of respiration parameters within the epoch. We can observe from the waveforms that most channels can get similar BR = 35, but the channels with smaller *σ_BR_* and *σ_PP_*, such as ‘gyro X’ and ‘gyro Y’, have more stable respiratory waveforms. Therefore, we opted to use the optimal channel by the minimum mean of all covariation features in BR, PP, IN, EX, and IBI. In Figure 4, the optimal channel is ‘gyro Y’.

Signal quality cannot be guaranteed during the entire course of overnight recording because patients may have random motions lying on the bed or leave the bed for restroom visits. Various factors, such as ambient movement, might bring about noises to cause SNR degradation. The position of the wearable sensor on the patient clothing might sometimes move during long or deep breathing and brought further noise to the signal. Therefore, we opted to remove the epochs with low SNR by predetermined thresholds. We selected the threshold to be the mean of all covariation features, which should be smaller than 0.4. Table 2d provides the selection ratio for every dataset and the final cases we have collected after all the signal processing procedures. The datasets from Exp 1 and Exp 2 have higher quality because of the better-controlled lab environment during data collection.

## 3. Results

### 3.1. Feature Analysis and Comparison

This study involved three distinct datasets: COVID-19 patients and two dyspnea human studies on healthy subjects, as summarized in Table 1. After acquiring all datasets, we first evaluated the similarity of respiratory features between COVID-19 patients and healthy subjects (Exp 1) where dyspnea was physiologically induced by exercise. The changes in respiratory features can have correlation with important implications.

We first examined a few representative respiratory features and presented the scatter plots in Figure 5a,b from 3 datasets: (1) COVID-19 patients by accelerometers; (2) healthy subjects during normal breathing in Exp 1 by NCS; and (3) healthy subjects breathing after exertion in Exp 1 by NCS. Respiratory features collected from NCS and accelerometers in our human study have a high similarity, so the differences using two different sources were mainly determined by SNR considerations. We also presented similar scatter plots using the accelerometer in Exp1 and NCS in Exp 2 in Appendix A, which gave comparable observations. In Figure 5a, the X and Y axes represented respiratory features of autocorrelation of BR and PP between successive breaths *ℜ_BR_* and *ℜ_PP_*, while in Figure 5b, the X and Y axes represented the absolute difference and autocorrelation of adjacent IBI *ς_IBI_* and *ℜ_IBI_*. To better compare the feature distribution for different datasets, we used the Gaussian kernel smoothing function to estimate the returned probability density in the top and right lines. The dataset from Healthy Normal in Exp 1 had a much broader range of distribution compared with the other two datasets of COVID-19 and Healthy Exertion. For a better visual demonstration, we set the X-Y limits to only show all points from COVID-19 and Healthy Exertion; some points from Healthy Normal were out of range. We can observe that COVID-19 data had a much higher similarity of respiratory features to Healthy Exertion. In Figure 5a for *ℜ_BR_* and *ℜ_PP_*, and in Figure 5b for *ℜ_IBI_*, both COVID-19 and Healthy Exertion participants had higher values closer to one, indicating higher autocorrelation of neighboring breathing cycles. In Figure 5b, for *ς_IBI,_*, COVID-19 and Healthy Exertion groups were concentrated on smaller successive differences, while Healthy Normal had a broader spread and higher variation.

For a more comprehensive comparison of similarities in different datasets, we calculated the Kullback–Leibler (KL) divergence, which is also called relative entropy, as a type of statistical distance between two probability distributions [32]. KL divergence provides a measure of the similarity between two known discrete probability distributions, *P* and *Q*, where *P* often represents the true data distribution and *Q* represents the predicted data distribution. KL divergence can be calculated by:(2)$$K\left(PQ\right)=\sum_{i}P_{i}\log(\frac{P_{i}}{Q_{i}})$$

To overcome the biases from the different sample sizes and reduce the random noises in the two distributions, we first transformed our discrete datasets to smoothed continuous distributions using Gaussian kernels as the top and right lines in Figure 5a. After retrieving the probability density function, we resampled the distribution to 200 data points for each dataset. In Table 3a, we presented eight representative respiratory feature statistics. The range of values for the KL divergence outputs is [0, +∞). KL distances close to zero imply the two distributions are very similar regarding the feature tested. The larger the KL distance, the larger the dissimilarity, or divergence. In both Exp 1 and Exp 2 with healthy subjects, Healthy Normal had larger KL divergence compared with those of COVID-19 than Healthy Exertion. In Table 3b, we also examined the dissimilarity between NCS and accelerometer in the same experiment of Exp 1. The small KL divergence close to zero between the two sensors showed the similarity and interchangeability for waveform measurements.

Before employing the ML model for analysis, we first compared respiratory features of COVID-19 to other datasets by KL divergence to help evaluate their similarity. For most of the features of importance, the COVID-19 dataset had a smaller KL divergence compared with Healthy Exertion than that of Healthy Normal. Results in the next section from the ML model also match the KL analysis here.

### 3.2. Dyspnea Classification Model

We found a high similarity of respiratory features between COVID-19 patients and healthy subjects with exertional dyspnea. We adopted our previous dyspnea model derived from Exp 2 [26] on 32 healthy subjects as the training model to blind-test the COVID-19 patients and the healthy subjects with the same sensor setup in Exp 1.

By utilizing the ML model in [26], we can evaluate the dyspnea score from all respiratory features as a whole and extrapolate the applicability of our objective dyspnea reporting system to the present study. As it is impractical to ask patients frequently to self-report their dyspnea scores, our alternative respiratory score can be of high value to provide a symptomatic gauge for COVID-19 patients continuously, especially during inconvenient periods, such as during sleep or under treatment.

In our previous dyspnea study on healthy subjects, we obtained self-reported dyspnea scores from all subjects as the ground truth for reference and supervised training. However, in the clinical study on COVID-19 patients confirmed by chest imaging, we reported the respiratory scoring results through inference only. Since the COVID-19 dataset had no explicit ground truth from self-reported dyspnea except at the very beginning, we did not employ the k-fold cross-validation process on all datasets. Instead, we adopted our previous dyspnea study on healthy subjects (Exp 2) as the training set and then fixed the learning model. Within this previous study of Exp 2, we performed the processes of k-fold cross-validation (accuracy = 87.1%) as well as leave-one-subject-out cross-validation (accuracy = 86.6%) to justify our objective dyspnea scoring model.

The first model we built was the binary classification model, namely, normal = 0 and dyspnea = 1. We built and trained the model based on our previous dyspnea study (Exp 2) and then made direct deductions on COVID-19 patients by testing on all COVID-19 cases and the control datasets on healthy subjects (Exp 1). Table 3c presents detailed information on training and testing sets, including the datasets we used from different studies, the input cases to the model, and the subject and routine composition of all cases. Even though the training set of Exp 2 has more subjects, the case number was relatively smaller due to the short duration (5 min) of the routines. In the COVID-19 dataset (duration averaged about 14 h) and its blind control of Exp 1 (duration of 30 min), we were able to accumulate more usable epochs after signal processing. In the fixed training set, all normal breathing epochs were labeled as normal = 0, while all exertional breathing epochs were labeled as dyspnea = 1. By training on the dataset to build a dyspnea classifier, we can output the dyspnea classification results for all COVID-19 datasets and the control cases from Exp 1. We utilized the k-nearest neighbor (kNN) classifier as the model here with k = 40. Before feeding the dataset into the model, all features were preprocessed with a standard scaler for regularization by removing the mean and scaling to unit variance.

As Table 3d shows, almost all COVID-19 patients’ cases were classified as dyspnea, while 73.6% of exertional breathing cases of healthy subjects were classified as dyspnea. In comparison, only 4.24% of the normal breathing cases in healthy subjects were classified as dyspnea. The dyspnea classification results for COVID-19 patients further corroborated our hypothesis that COVID-19 respiratory features had a high similarity to exertional dyspnea features in healthy subjects. We also observed similar results using accelerometers in Exp 1 in Appendix A.

Apart from the epoch-wise binary classification on the COVID-19 dataset, we also presented subject-wise results, as each subject had a self-reported dyspnea interview at the hospital admission, which is presented in Appendix A with detailed information. We first calculated the epoch-wise classification results of dyspnea or normal from all epochs and then averaged for a final result for every patient. The class labels were the rounded values of the average. In Figure 6, we present the subject-wise confusion matrix of deduced dyspnea vs. self-reported dyspnea. Control cases in Exp 1 are included in Figure 6a, while only COVID-19 cases are included in Figure 6b. Notice that Case 5 of COVID-19 patients reported no dyspnea at the beginning but later reported that he/she became more dyspneic after coughing precipitated by speaking. The uncertainty and subjective nature of self-reporting can contribute to lower accuracy in our respiratory analysis.

### 3.3. Correlated Respiratory Scores

In this section, we employed a regressor model for “correlated respiratory scores” in the Borg scale (0–10) *D_corr_* for COVID-19 patients and the control group. In our previous work, we built a similar objective dyspnea score *D_obj_* [24], which was reasonably accurate for exertional dyspnea on healthy subjects in comparison with self-reported *D_self_*. In this study, we used our previous dyspnea study as the training model to build the correlated respiratory score *D_corr_* and treated the data from COVID-19 patients and Exp 1 as blind testing cases. We implemented the kNN regressor as the main model here. Since we had overnight recording for COVID-19 patients, we first calculated the epoch-wise *D_corr_* from all datasets and then averaged all *D_corr_* as the final score for every patient.

Figure 7a presents the results for *D_corr_* of COVID-19 patients and healthy subjects in Exp 1. For average *D_corr_* extracted from different datasets, COVID-19 = 4.39; Healthy Normal = 1.26; Healthy Exertion = 2.72. As observed from Figure 7a, *D_corr_* for COVID-19 was more concentrated between 4 and 5, while *D_corr_* for Healthy Normal was mostly below 2. Healthy Exertion had more participants with higher *D_corr_*, but subject variation was also evident, possibly because different participants had variations in physical conditioning after the same cardio exercise. *D_corr_* for COVID-19 data was less dispersed probably due to the more uniform manifestations of the underlying pulmonary involvement. We also presented similar results using the accelerometer in Exp 1, as shown in Appendix A.

We further performed T-tests for *D_corr_* results on different datasets as shown in Table 3e. The calculated T-statistic is positive when the sample mean of the first dataset is greater than that of the second dataset, and negative otherwise. As the T-statistic showed, *D_corr_* for COVID-19 was distinctively higher than for Healthy Normal, while the difference with Healthy Exertion was smaller. The very small *p*-value between COVID-19 patients and Healthy Normal indicated that they had distinctively different *D_corr_* distributions. For the Healthy Exertion data, the sufficiently small *p*-values for COVID-19 and Healthy Normal suggested high distinguishability between the datasets.

Apart from reporting average *D_corr_* to give a subject-wise evaluation, our system can also output continuous real-time *D_corr_* to indicate the progression of lung functions. COVID-19 and other pulmonary diseases, such as COPD, are frequently accompanied by dyspnea sensations from reduced lung function. It is thus critical to continuously monitor patients because the infection often develops insidiously over a period of time. The frequent self-reporting is inconvenient and less accurate for long-term tracing of the symptoms. Figure 7b shows an example of continuous monitoring of *D_corr_* for six COVID-19 patients. In the whole recording of 12–16 h, *D_corr_* was reported every hour, to align with the clinical data recorded every hour.

## 4. Discussion

In this paper, we have tried to construct a “correlated respiratory score” based on measured respiratory features, as a surrogate analog to the self-reported dyspnea score. As dyspnea is presently defined as the subjective experience of breathing discomfort, the correlated respiratory score can at best represent the physiological origin of the dyspnea sensation, but its continuous, objective, and autonomous evaluation should be a useful supplement to the self-reporting.

Challenges to constructing a clinically acceptable respiratory evaluation platform still remain and will need to be addressed in future studies:Presently we only deduced respiratory scores of COVID-19 patients using our previous ML model built on physiologically induced dyspnea on healthy subjects. We had limited ground truth about the dyspnea experienced by COVID-19 patients during the long-term continuous monitoring. In the future, we should gather more clinical information that can help evaluate lung involvement, even if not collected repetitiously. For example, continuous measurements of oxygen saturation (SpO2) and heart rates can be used as indirect references to respiratory conditions.SNR of the present wearable sensors in clinical settings needed further improvement. We relied on epoch selection to eliminate noisy periods. Sensor improvement for higher SNR and higher tolerance to subject motion interference should be investigated.The population size for this study was relatively small (n = 12). In the future, we should broaden the demographic diversity of the clinical studies to be more statistically significant, with probable inclusion of pneumonia patients from different disorders.We can extend the study to patients with various cardiopulmonary disorders of lung involvement to establish the true effectiveness of the proposed respiratory score, for example, COPD, asthma, and other types of pneumonia.Feature selection and reduction is still desirable in future studies. To avoid the “curse of dimensionality” and to find the most correlated respiratory features, we can implement feature reduction algorithms before inputting data into the ML model.

## 5. Conclusions

Dyspnea is a key symptom for patients with COVID-19 and many other respiratory disorders. Existing clinical evaluation of dyspnea currently depends on self-reporting, which is subjective and inconvenient for continuous monitoring and for patients with compromised communication abilities. In this paper, we propose an innovative approach to continuously monitor respiratory features using wireless and wearable sensors to develop a correlated respiratory scoring system deduced from exertion routines on healthy subjects. COVID-19 patients could hardly provide a self-report of dyspnea in a continuous and long-term manner, especially over the rest period during the night time. Hence, we provided the deduced respiratory scores on COVID-19 patients and control subjects under the same sensor setup by learning from our previously validated dyspnea model. We found a high similarity between COVID-19 patients and healthy subjects after exertion, suggesting that changes in respiratory features from physical exertions could be analogous to those from pulmonary disorders. We also demonstrated the unique capability to continuously report correlated respiratory scores *D_corr_* during 12–16 h for COVID-19 patients. Our system can be a viable tool to help with diagnosis and prognosis of COVID-19, offering a warning of possible worsening respiratory function, as well as evaluation of the degree of recovery. This work can be potentially applied to symptomatic evaluation of patients with similar respiratory conditions including asthma, pneumonia [33], and COPD [34].

## Figures and Tables

**Figure 1 sensors-23-04733-f001:**
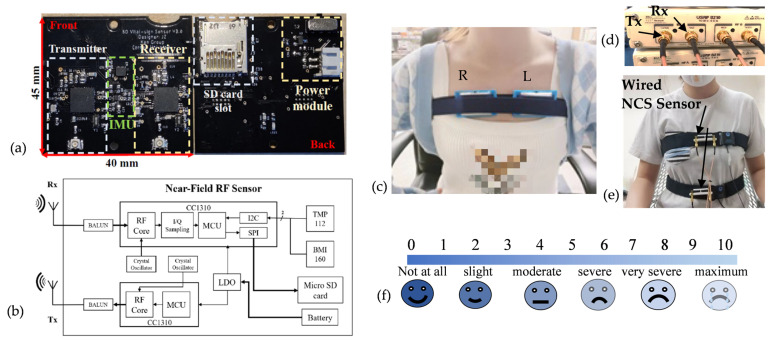
(**a**,**b**) The all-in-one wireless wearable respiratory sensor on PCB: (**a**) the front and back photos; (**b**) the block diagram for the sensor unit. (**c**) Body deployment of two wireless units by a chest belt; (**d**) SDR transceivers used for the RF sensors. (**e**) Wired RF sensors that were connected by cables to the sensing antennas on the chest and abdomen of a participant. (**f**) Description of the self-reported Borg scale for dyspnea.

**Figure 2 sensors-23-04733-f002:**
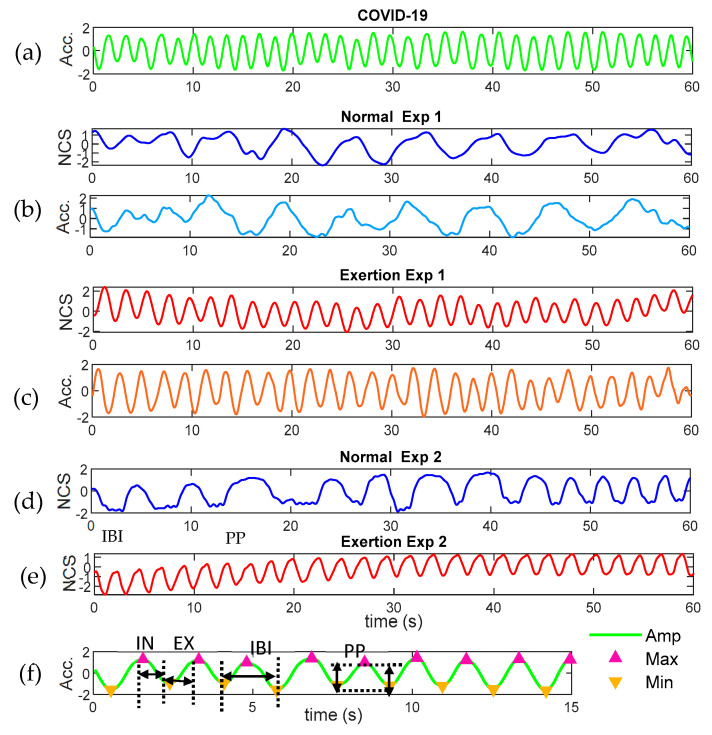
Waveform examples: (**a**) COVID-19 patients; (**b**) healthy normal breathing in Exp 1; (**c**) healthy post-exertion breathing in Exp 1; (**d**) healthy normal breathing in Exp 2; (**e**) healthy post-exertion breathing in Exp 2; (**f**) min-max peak detection for respiratory parameter extraction.

**Figure 3 sensors-23-04733-f003:**
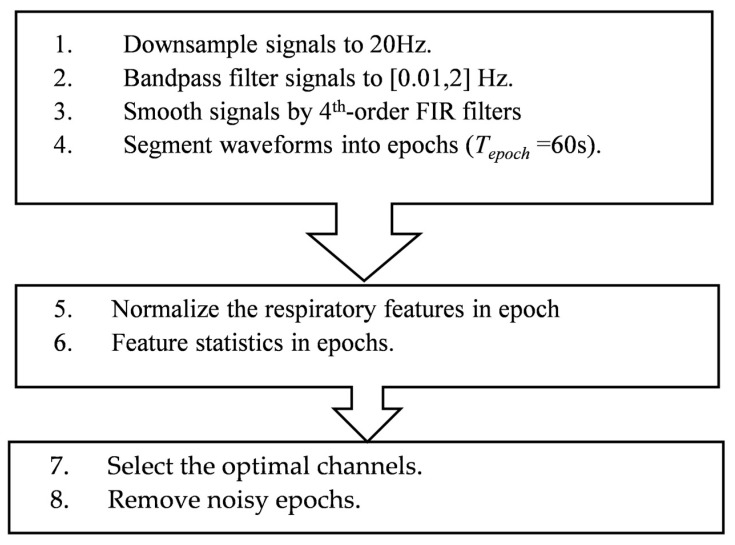
Signal processing procedures of respiratory datasets.

**Figure 4 sensors-23-04733-f004:**
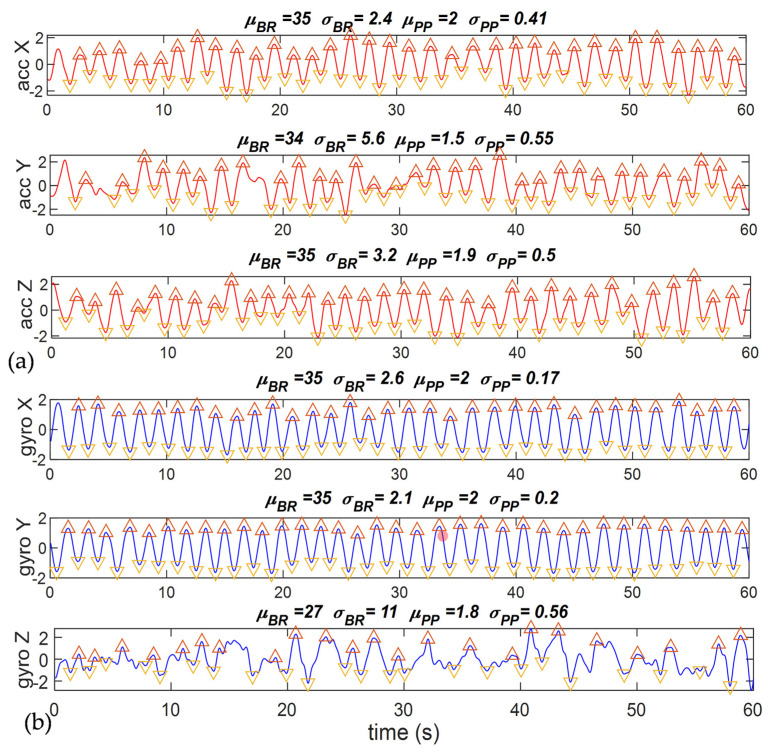
An example of channel selection for the accelerometer. The optimal channel is ‘gyro Y’. (**a**) Accelerometers in (X, Y, Z) axes; (**b**) gyroscopes in (X, Y, Z) axes.

**Figure 5 sensors-23-04733-f005:**
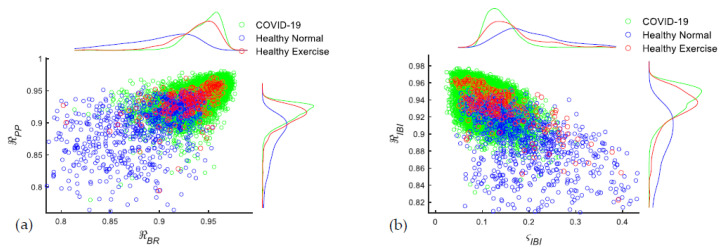
(**a**,**b**) Scatter plots of chosen respiratory features from COVID-19 and Exp 1 datasets. Top and right lines are smoothed continuous distribution by Gaussian kernels. (**a**): *ℜ_BR_* and *ℜ_PP_*; (**b**): *ς_IBI_* and *ℜ_IBI_*.

**Figure 6 sensors-23-04733-f006:**
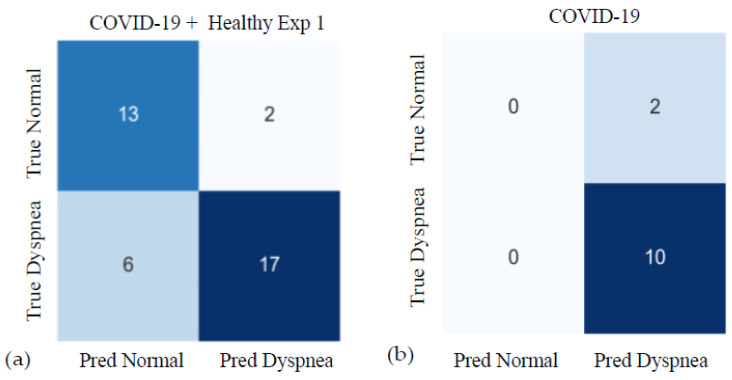
Confusion matrices for subject-wise dyspnea classification results. The cells list the number of subjects in each category. (**a**) Combined results from COVID-19 patients and the control healthy subjects in Exp 1. Accuracy = 0.79; sensitivity = 0.74; specificity = 0.87. (**b**) Results from COVID-19 patients only. “True” Dyspnea/Normal are based on self-reported dyspnea at rest at the hospital admission.

**Figure 7 sensors-23-04733-f007:**
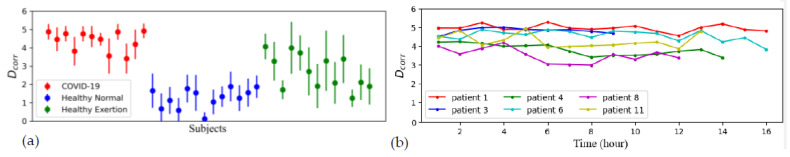
(**a**) Respiratory scoring results for COVID-19 patients and healthy subjects in Exp 1. The average of *D_corr_*: COVID-19 = 4.39; Healthy Normal = 1.26; Healthy Exertion = 2.72. Dots and bars represent mean and standard deviations of *D_corr_* for each subject. (**b**) An example of continuous monitoring of correlated respiratory scores for COVID-19 patients for every hour.

**Table 1 sensors-23-04733-t001:** (**a**) Demographics in human study. (**b**) Acquisition of different datasets.

(a)				
Datasets	Gender	Number	BMI (µ ± σ)	Age (µ ± σ)
COVID-19	Male	8	30 ± 7.3	-
	Female	4	28 ± 6.3	-
Exp 1	Male	7	23 ± 2.5	29 ± 12
	Female	6	21 ± 3.7	21 ± 2
Exp 2	Male	14	23 ± 2.5	28 ± 9
	Female	18	20 ± 1.3	24 ± 2
**(b)**				
	**Participants**	**Recording Time**	**Sensors**	
COVID-19	12 COVID-19 patients	Continuous 14 h	NCS with accelerometers	
Exp. 1	13 healthy subjects	1. Normal (30 min) 2. Post-exercise (5 min)	NCS with accelerometers	
Exp. 2 [26]	32 healthy subjects	1. Normal (5 min) 2. Post-exercise (5 min)	Wearable NCS by SDR	

**Table 2 sensors-23-04733-t002:** (a) Instantaneous respiratory parameters (7). (b) Respiratory features (37). (c) Frequency features (10). (d) Statistical comparison of COVID-19 and different datasets.

(a)							
Extracted Parameters				Description			
Breath Rate (*BR*)				Inverse of the interval between two neighboring minima			
Peak-to-Peak (*PP*)				Lung volume estimated by difference in successive peaks			
Inhalation Interval (*IN*)				Time difference between a minimum and the next maximum			
Exhalation Interval (*EX*)				Time difference between a maximum and the next minimum			
Inter-Breath Interval (*IBI*)				Interval between two neighboring maxima			
In- Ex Ratio (*IER*)				Inhalation/exhalation interval ratio			
In- Ex Volume Ratio (*IEPP*)				Inhalation/exhalation volume ratio			
**(b)**							
*µ_BR_*	*µ_PP_*	*µ_IN_*		*µ_EX_*	*µ_IBI_*	*µ_IER_*	*µ_IEPP_*
*σ_BR_*	*σ_PP_*	*σ_IN_*		*σ_EX_*	*σ_IBI_*	*σ_IER_*	*σ_IEPP_*
*CoV_BR_*	*CoV_PP_*	*CoV_IN_*		*CoV_EX_*	*CoV_IBI_*		
*ℜ_BR_*	*ℜ_PP_*	*ℜ_IN_*		*ℜ_EX_*	*ℜ_IBI_*	*ℜ_IER_*	*ℜ_IEPP_*
*ς_BR_*	*ς_PP_*	*ς_IN_*		*ς_EX_*	*ς_IBI_*	*ς_IER_*	*ς_IEPP_*
*µ_skew_*	*µ_kurt_*	*entropy*		*cycle*			
**(c)**							
*ƞ_f1_*	*ƞ_f2_*			*ƞ_f3_*		*ƞ_f4_*	
*ƿ_f1_*	*ƿ_f2_*			*ƿ_f3_*		*ƿ_f4_*	
*f_BR_*	*SNR_BR_*						
**(d)**							
	**COVID-19** **Acc.**	**Norm.** **NCS Exp 1**	**Exer.** **NCS Exp 1**	**Norm.** **Acc.** **Exp 1**	**Exer.** **Acc.** **Exp 1**	**Norm.** **NCS** **Exp 2**	**Exer.** **NCS** **Exp 2**
Cases	10,131	1049	188	918	231	256	240
Ratio (%)	30.2	74.0	77.7	64.7	95.5	100	100

**Table 3 sensors-23-04733-t003:** (a) KL-Divergence of COVID-19 to other datasets. (b) KL-Divergence of NCS and accelerometers in Exp1. (c) Training/testing sets in the model. (d) Classification results of dyspnea for COVID-19 patients and healthy subjects in Exp 1. (e) T tests for correlated respiratory scoring.

(a)						
	Norm NCS Exp 1	Exer NCS Exp 1	Norm ACC Exp 1	Exer ACC Exp 1	Norm NCS Exp 2	Exer NCS Exp 2
*µ_BR_*	2.14	0.17	2.62	0.16	1.44	0.30
*σ_BR_*	1.71	0.69	0.68	0.72	0.49	0.91
*CoV_BR_*	3.91	0.76	2.75	0.65	1.17	1.03
*CoV_IBI_*	4.79	0.99	3.37	1.08	1.66	1.33
*ℜ_BR_*	3.42	0.16	2.97	0.16	1.52	0.50
*ℜ_PP_*	2.96	0.21	2.34	0.08	0.98	0.21
*ς_IBI_*	3.87	0.44	2.61	0.68	1.32	0.73
*ς_IER_*	1.82	0.26	1.34	0.16	0.43	0.17
Avg	3.08	0.46	2.34	0.46	1.13	0.65
**(b)**						
	** *µ_BR_* **	** *σ_BR_* **	** *CoV_BR_* **		** *CoV_IBI_* **	
Norm	0.05	0.09	0.12		0.13	
Exer	0.01	0.13	0.11		0.06	
	*ℜ_BR_*	*ℜ_PP_*	*ς_IBI_*		*ς_IER_*	Avg.
Norm	0.04	0.05	0.12		0.21	0.10
Exer	0.08	0.04	0.22		0.08	0.09
**(c)**						
	**Training Set:**		**Testing Set:**			
	**Healthy Normal Exp 2**	**Healthy Exertion** **Exp 2**	**COVID-19**		**Healthy Normal Exp 1**	**Healthy Exertion** **Exp 1**
Cases	256	240	10,131		1049	188
Subjects	32	32	12		13	13
Routine /subject	5 min	5 min	~14 h		30 min	5 min
**(d)**						
			**COVID-19**		**Healthy Normal**	**Healthy Exertion**
Percentage of Dyspnea			98.05%		4.24%	73.63%
**(e)**						
	**COVID-19 vs.** **Healthy Normal**		**COVID-19 vs.** **Healthy Exertion**		**Healthy Normal vs. Healthy Exertion**	
T-statistic	14.60		5.47		−4.82	
*p*-value	4.61 × 10^−13^		2.75 × 10^−5^		1.1 × 10^−4^	

## Data Availability

The collected and analyzed datasets during the current study are available from the corresponding author upon request.

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
