# Peer review of "Deduced Respiratory Scores on COVID-19 Patients Learning from Exertion-Induced Dyspnea"

_sensors, 2023, doi:10.3390/s23104733_

Round 1

Reviewer 1 Report

The aim of this study was to determine whether dyspnea in COVID-19 patients can be assessed using a wearable sensor and whether a correlated respiratory score can be established from a learning model based on physiologically induced dyspnea on healthy subjects. This study is interesting. I have reviewed this paper and commented as follows:

1-       There are some typographical errors in this present manuscript, Such as COVID-19 or COVID. Would the authors please review this manuscript carefully again. COVID-19 should be given full spelling in first place.

2-       In the section of Results, are the authors sure that “3.1 Physiological aspects of dyspnea” were the results obtained from this present study. In addition, the methods should not be described in the section of Results.

3-       In the section of Discussion, would the authors please address the limitations of the present study and further improvement.

4-       Why did the authors not compare the present methods for dysnea detection with conventional and gold standard diagnosis for dysnea in clinical practice?

Reviewer 2 Report

I think this is an interesting research question and the authors put great efforts to conduct this study and analyze data. However, frankly speaking, the paper is sometimes difficult to read and I think there a some serious methodological drawbacks which should be improved, if possible: 

1. Tab 3 (c). Please improve the readability of this table by providing sensitivity, specificity, negative, positive predictive power and accuracy for the classification of dyspnea in terms of percentages and absolute numbers.

Additionally, to increase transparency, I suggest to make a cross-tabulation table with rows which correspond to the observed numbers of dyspnea  versus the predicted numbers of dyspnea (columns) in terms of absolute numbers and percentages. 

Please add 95% confidence intervals to sensitivity, specificity, negative, etc. 

2. Most importantly, please describe whether results in Table 3c are based on the whole data set or based on the training sample alone or an independent test sample. This is an important information for the reader because it is important to analyze how the machine learning model performed in a training sample and how it performed in the independent test sample. This is very important to demonstrate that the ML-model does not suffer of overfitting and generalizes well to new, previously unseen data. If this is not done, it is not statistically sound to claim that a model is somehow validated.

I think it is crucial to extensively report the performance of a machine learning model in terms of sensitivity, specificity, negative, positive predictive power and accuracy in a training and to compare it with results in an independent test sample. Why this? Because it is crucial to know how the model performs when confronted with new previously unseen patients.

Unfortunately, there are too many papers published which were not sufficiently tested, but the authors claim that their model works, but unfortunately fail when confronted with new data.

This is very important to clarify this point. Only if this was carefully done, it can be claimed that the model works (see also line 22: "This paper validates the viability to use our correlated respiratory scoring 22 for clinical dyspnea assessment on COVID patients")

3. In line 189 you mention that epoch segmentation of the waveforms were done. So, if we take a look at the waveform of one single patient: Does this mean you split this waveforms into several epochs (pieces) and then you fed all these pieces into the knn-classifier? 

If this is done in this way, what I suspect, the model cannot be used. Because this would artificially increase the sample size from a dozen of patients to several hundreds or even thousends of such pieces.

From a statistical point of view, this is inacceptable as any statistical sample has to be sampled independently! If e.g. from one patients hundreds or thousends such pieces were selected, the independence of the sample is no longer guaranteed and no machine learning model can be applied. 

Please describe step by step how which data were fed for training of the model.

4. Did you consider to use n-fold cross-validation to train the machine learning model? As far as I can see, not. Why did not used it? Due to the small training sets? What were the reasons?

I recommend to use n-fold crossvalidation to get more stable results.

5. Results section: I wonder a bit why the results start with "Physiological aspects of dyspnea". I think these are more general medical statements and nothing new for a clinician? Please argue why this subsection is the first part of the result section.

6. Please add the absolute numbers and percentages of probands suffering of dyspnea? Did all COVID patients suffer from dyspnea? 

7. You used k-nearest neighbor classifier. Did you also consider to use recurrent neural networks? May be this would simplify feature selection, because the whole waveform can be used as input vector. Recurrent neural networks might improve model performance.

8. In the paper, several times it was claimed that your approach was validated. Unless, the models or statements were not independently tested, it is not possible to claim such statements. The danger that the model does not perform with the same accuracies is too high unless it is tested in an independently test sample.

9. Line 296: you write: "For a more comprehensive comparison of similarity in different datasets, we calculated the Kullback–Leibler (KL) divergence, which is also called the relative entropy as a type of statistical distance between two probability distributions."

I think Kullback-Leibler divergence is quite difficult to interpret for most readers. For example, how can a KL-divergence value of 0.05 be interpreted when compared to a value of e.g 0.10.

I cannot see the necessary to present KL divergence to describe similarity of different distributions. If you think this is necessary, please argue why you think this is necessary.

10. Tab. 3 (d). I think the study was done to finally apply respiratory scoring on an individual/personalized level. Comparing pairwise means between three groups is nice, but I suggest to present 3 histograms of the empirical respiratory scores in one diagram for COVID, healthy normals and healthy exertion patients.

I think this might be useful such that the reader gets a better impression of the overlap between these 3 empirical distributions. 

11. For model training, did you use any feature selection algorithm to further reduce the set of predictor variables, such as a genetic algorithm for feature selection?

12. Line 310-312: You write “by utilizing the ML model in [24], we can evaluate the dyspnea score from all respiratory features as a whole and validate the applicability of your objective dyspnea reporting system in the clinical setting”. This is not clear to me what this actually means. Please describe step by step in a transparent and easy way what was exactly done at each step.

I think the quality of English language is good. However, sometimes I think the manuscript is difficult to read. 

To increase the readability for the reader, I recommend to add a subsection in the statistical methods how the machine learning model was trained and tested: What was the training sample? How large was it (n=?) Were the waveforms split up into epochs and were these pieces fed into the knn-classifier?  How large was the training sample (n=?). Was  the knn-classifier tested? If so, how large was the training sample? 

Please add a subsection how model performance was analyzed using sensitivity, specificity, neg. and pos. pred. power, total accuracy.

Round 2

Reviewer 1 Report

Thanks a lot for the authors' anwsers. I have no further questions.

Author Response

NA

Reviewer 2 Report

Dear authors,

thank you for your replies. However, still I have a lots of questions which I think should be clarified.

On the one hand, you explained that "The main logic behind this paper is to deduce the pneumonia-induced dyspnea in COVID patients using exertional dyspnea in healthy subjects. Our main conclusion in this work is that an objective and continuous evaluation system for dyspnea can be achieved through respiratory feature comparison with the exertion-induced dyspnea model."

On the other hand, you explained that the ground truth, i.e. whether or not a Covid patient suffered from dyspnea is missing. 

I do not see the benefit of a system for dyspnea evaluation, if it is not even known if a patient suffered from dyspnea?? 

In the first version of the manuscript you claimed: ""This paper validates  (!) the viability to use our correlated respiratory scoring for clinical dyspnea assessment on COVID patients" -

and now you write "This study offers a useful system for dyspnea evaluation on patients with active or chronic respiratory disorders, especially towards the patient population who refuse to cooperate  or cannot communicate due to deterioration or loss of cognitive functions, where the dyspnea evaluation model has to be achieved through inference as individual self-report cannot be available"

How is this "usefulness of this system" be guaranteed? Please explain!

Only because they had high respiratory scores? How do you decide whether or not a given subject suffers from dyspnea? If you are confronted with a Covid patient and you get a deduced respiratory score say x, what are you doing with the value x? Is there cut-off for saying: if x is larger than a certain value, then the patient suffers of dyspnea? What are you doing with this value? Please explain.

Please note that you wrote in the abstract: "This study is to determine if dyspnea in COVID-19 patients can be assessed using a wearable sensor and if a deduced respiratory score can be established from a learning model based on physiologically induced dyspnea on healthy subjects."

I think this is a large discrepancy between what your deduced system does in reality - in fact providing an respiratory score - and what you claim (!). You found a similarity in the distribution, that is fine. But this is not enough. I want to see an accordance between what the model claims and reality. 

May be your system was confronted with Covid patients without having dyspnea, but the system deduces that a certain Covid patient has a high respiratory score, but in fact, he or she does not have dyspnea?

You mentioned that "We found high similarity between COVID-19 patients and healthy subjects after exertion, suggesting that changes of respiratory features from physical exertions could be representative of the dyspnea found in pulmonary disorders." (line 429-431).

But "similarities" are not enough to claim to suggest that this the system is "useful". It is only useful if it was proven (!) to be useful and if this is clearly stated what is meant with "usefulness". A high deduced respiratory score alone is not sufficient. 

I think the study would greatly benefit, if you provide how you deduced system performs in an independent test sample. Fig. S3 is also confusing: First you describe that you could not assess whether or not a Covid patient suffered of dyspnea and now you provide "classification results". 

Please make a table, observed vs predicted - if possible.

In fact, I think you your deduced system makes sense, but I ask you  to provide more evidence that it works when confronted with reality. I ask you also to provide good reasons for the usefulness of the model. 
